# Single-Beam Acoustic Doppler Profiler and Co-Located Acoustic Doppler Velocimeter Flow Velocity Data

**Marilou Jourdain de Thieulloy** [1],* , **Mairi Dorward** [1] , **Chris Old** [1] , **Roman Gabl** [2] , **Thomas Davey** [2] , **David M. Ingram** [2] and **Brian G. Sellar** [1]

1    School of Engineering, Institute for Energy Systems, The University of Edinburgh, Max Born Crescent, Edinburgh EH9 3BF, UK; m.dorward@ed.ac.uk (M.D.); C.Old@ed.ac.uk (C.O.); brian.sellar@ed.ac.uk (B.G.S.)
2    School of Engineering, Institute for Energy Systems, FloWave Ocean Energy Research Facility, The University of Edinburgh, Max Born Crescent, Edinburgh EH9 3BF, UK; roman.gabl@ed.ac.uk (R.G.); tom.davey@flowave.ed.ac.uk (T.D.); david.ingram@ed.ac.uk (D.M.I.)
*    Correspondence: marilou.jourdain@ed.ac.uk; Tel.: +44-0-131-6505646

**Abstract:** Acoustic Doppler Profilers (ADPs) are routinely used to measure flow velocity in the ocean, enabling multi-points measurement along a profile while Acoustic Doppler Velocimeters (ADVs) are laboratory instruments that provide very precise point velocity measurement. The experimental set-up allows laboratory comparison of measurement from these two instruments. Simultaneous multi-point measurements of velocity along the horizontal tank profile from Single-Beam Acoustic Doppler Profiler (SB-ADP) were compared against multiple co-located point measurements from an ADV. Measurements were performed in the FloWave Ocean Energy Research Facility at the University of Edinburgh at flow velocities between 0.6 ms$^{-1}$ and 1.2 ms$^{-1}$. This paper describes the data; the analysis of the inter-instrument comparison is presented in an associated Sensors paper by the same authors. This data-set contains (a) time series of raw SB-ADP uni-directional velocity measurements along a 10 m tank profile binned into 54 measurements cells and (b) ADV point measurements of three-directional velocity time series recorded in beam coordinates at selected locations along the profile. Associated with the data are instrument generated quality data, metadata and user-derived quality flags. An analysis of the quality of SB-ADP data along the profile is presented. This data-set provides multiple contemporaneous velocity measurements along the tank profile, relevant for correlation statistics, length-scale calculations and validation of numerical models simulating flow hydrodynamics in circular test facilities.

**Dataset:** https://doi.org/10.7488/ds/2842.

**Keywords:** offshore renewable energy; tank testing; acoustic doppler profiling; tidal current

---

## 1. Introduction

Knowledge of flow hydrodynamics in test tanks is critical for multiple tank testing applications. These include the testing at scale of offshore renewable energy (ORE) devices [1,2] and their sub-systems, as well as related applications such as the development and testing of underwater vehicles [3]. In terms of ORE, measurement of flow over a wide area of the tank is especially important for tidal stream energy where flow conditions directly influence turbine loads and generated power [4,5].

Flow measurement in test tanks is typically performed using point measurement techniques such as Laser Doppler Velocimetry (LDV) or Acoustic Doppler Velocimetry (ADV) [6–10]. Doppler technology based instruments estimate velocity using the Doppler shift in frequency between transmitted and return signals which results from back scattering by water-borne particulates. The Doppler shift between the transmitted and received pulse depends on the flow velocity, assuming that the particulates are moving at the same speed as the water. In the test tank environment, the scatterers comprise glass micro-beads with neutral buoyancy. ADV and LDV measure velocity by utilising multiple non co-planar beams (acoustic and light respectively) that converge and intersect at a small volume, the location of the velocity measurement. Distributed measurements can only be obtained from point measurement instruments by deploying multiple sensors or re-locating the device and repeating flow conditions, both of which are time and resource intensive.

Acoustic Doppler Profilers (ADP) capture multiple, virtually simultaneous, measurements along a profile and were originally designed for operation in environments such as open oceans, coastal waters and rivers. ADPs typically comprise multiple transducers mounted in a divergent configuration. Each transducer transmits and receives acoustic pulses to measure radial velocities along each acoustic beam. Recent increases in the carrier and sampling frequencies of ADPs has enhanced their potential for use at tank scales [11,12].

In this paper we present data from the inter-instrument comparisons and tank-testing work of Jourdain de Thieulloy et al. [13], providing further detail on experimental setup and acquired measurements. In Jourdain de Thieulloy et al. [13], a profiling instrument was introduced in the context of tank testing, namely a Nortek 1 MHz single-beam acoustic Doppler profiler (SB-ADP). The single-beam ADP described herein provides a uni-directional velocity component measurement. The use of a single beam reduces insonification of the tank outwith the zone of interest, reduces beam reflection at the tank sides, floor and water surface and eliminates measurement uncertainty associated with divergent beams.

This paper reports data associated with the velocity profiles measured by the SB-ADP and the co-located ADV velocity measurements at a range of typical tank operating velocities. The complete analysis and inter-instrument comparison is presented in the associated paper Jourdain de Thieulloy et al. [13]. The associated data set is available in the DataShare repository of the University of Edinburgh [14]. The data set contains SB-ADP data for nominal tank flow velocities of $0.6\ \text{ms}^{-1}$, $0.8\ \text{ms}^{-1}$, $1.0\ \text{ms}^{-1}$ and $1.2\ \text{ms}^{-1}$, and ADV data acquired at four locations along the profile for tank nominal tank flow velocities of $0.6\ \text{ms}^{-1}$, $1.0\ \text{ms}^{-1}$ and $1.2\ \text{ms}^{-1}$. ADV point measurements at nominal tank flow speed of $0.8\ \text{ms}^{-1}$ were collected during the SuperGen Marine tidal array project [5] and are available in Reference [15]. The SB-ADP configuration generated 54 simultaneous measurements of uni-directional velocity over a 10 m profile.

## 2. Material and Methods

The Nortek SB-ADP (Figure 1), is a variant of the Nortek Signature 1000 AD2CP [16]. SB-ADP measurements were compared against a Nortek Vectrino Profiler ADV [17] configured in non profiling mode. The component of flow velocity captured by the SB-ADP is the radial flow velocity along the acoustic beam formed by the sound waves. In the configuration presented in Figure 2 the SB-ADP acoustic beam points into the flow, directly measuring the stream-wise flow velocity component $u$. The profile is binned into a number of along-beam range cells by time-gating the reflected acoustic signal [18]. Hence, the SB-ADP can provide next to simultaneous velocity measurement of the $u$-component of the flow, reporting a velocity in each measurement cell along the profile.

The ADV provides a measurement of 3-dimensional velocity over a small sample volume $(1–4.5\ \text{cm}^3)$. Instrument specifications are contained within References [19,20]. Table 1 details the instruments settings utilised during the experiments with further detail in the metadata (Section 4). Calibration of both instruments was performed by the manufacturer and certificates supplied.

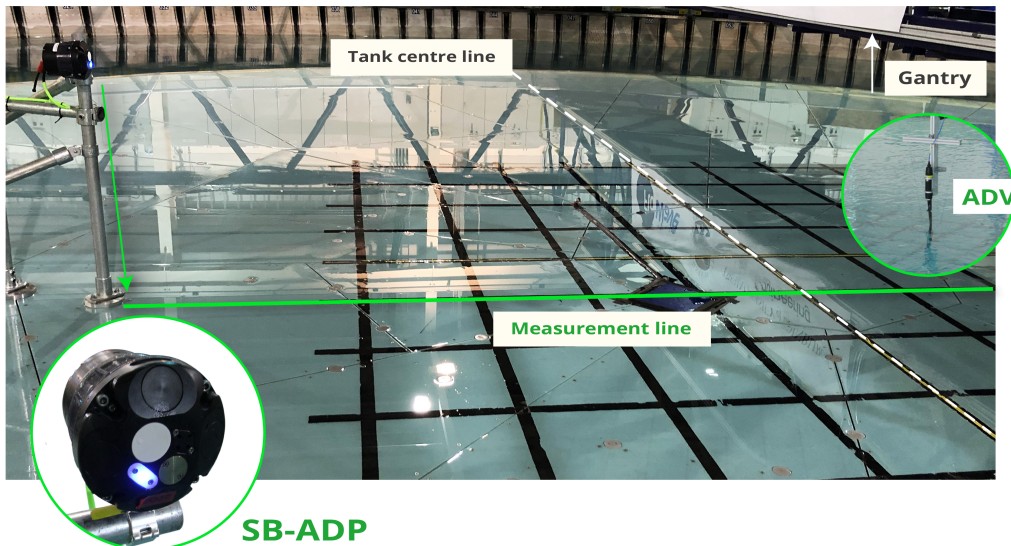

**Figure 1.** Single-Beam Acoustic Doppler Profiler (SB-ADP) set-up with the tank floor lifted up.

**Table 1.** Instruments characteristics during experiments.

| Instrument Type | Model | Abbreviation | Operating Frequency (MHz) | Sample Rate (Hz) | Cell Size (cm) |
|---|---|---|---|---|---|
| Acoustic Doppler Profiler | Nortek Single-beam | SB-ADP | 1 | 16 | 20 |
| Velocimeter | Nortek Vectrino Profiler | ADV | 10 | 100 | 0.4 |

## 2.1. Experimental Set-Up

The test set-up schematic is illustrated in Figure 2. A Cartesian coordinate system is used to locate the instruments in the tank, with origin at the tank centre on the floor $((x_t, y_t, z_t) = (0, 0, 0))$, $z_t$ vertically upwards and $x_t$ pointing into the main flow direction). Instrument coordinates with reference to the tank centre are provided in Table 2. A translated coordinate system $((x, y, z))$, originating at the SB-ADP transducer is used to facilitate analysis of the velocity measurements along the profile.

The SB-ADP was mounted on a floor bolted stand with sufficient structure stiffness to avoid vibration of the system. Laser alignment verified that the acoustic beam lay parallel to $x_t$ of the tank coordinate system. The SB-ADP theoretical acoustic beam is represented in Figure 2 as a cone of 2.9° width, binned into measurement cells.

The ADV was mounted onto the gantry which can be moved along the $x_t$-axis of the tank to capture successive point measurements within the sample volumes of the SB-ADP. The gantry positions are recorded with millimetre precision.

The ADV flexible head was used to enable sampling at the same location in the $z$-axis as the SB-ADP. The flexible head was mounted on a rigid beam with a kite structure to minimise vibration. The ADV sample volume was aligned with the centre line of the SB-ADP beam. As the ADV head and its installation elements were acting as a hard stationary scatterer, measurements were conducted separately and the ADV was removed from the water while the SB-ADP was measuring.

ADV locations along the profile are displayed in Table 3 with reference to both the tank centre and the SB-ADP. Note that ADV data at multiple locations within the tank are available in Reference [15].

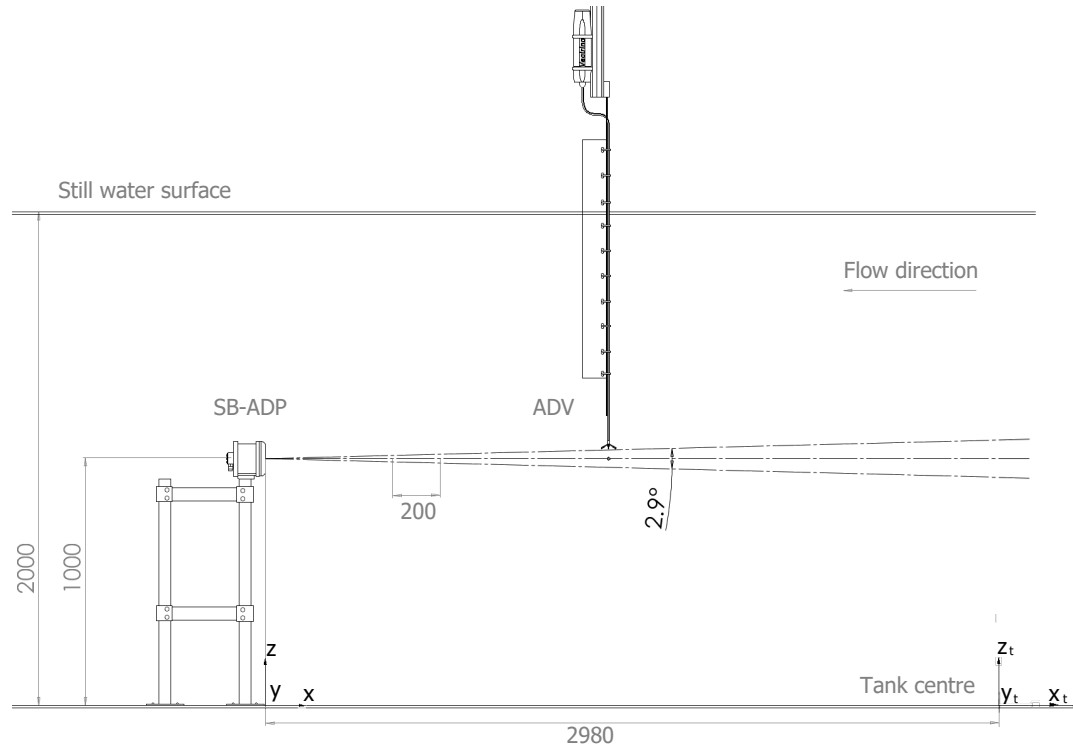

**Figure 2.** Diagram of the test set-up showing the SB-ADP pointed into the flow, and its theoretical acoustic beam binned into measurement cells. The Acoustic Doppler Velocimetry (ADV) is mounted on an *x*-adjustable gantry, enabling ADV point measurements to be made within successive SB-ADP measurement cells [13].

**Table 2.** Instrument coordinates with reference to the tank centre.

| Instrument | Coordinates (m) | | |
| --- | --- | --- | --- |
| | $x_t$ | $y_t$ | $z_t$ |
| **SB-ADP** | −2.98 | −0.545 | 1.00 |
| **ADV** | variable see Table 3 | −0.545 | 1.00 |

**Table 3.** ADV locations along the profile with reference to the tank centre ($x_t$) and to the SB-ADP ($x$).

| | $x_t$ (m) | $x$ (m) |
| --- | --- | --- |
| **A** | −2.181 | 0.799 |
| **B** | −0.774 | 2.206 |
| **C** | 1.019 | 3.999 |
| **D** | 3.013 | 5.993 |

*2.2. Test Conditions*

Measurements were conducted individually. Repeatability of test conditions within the tank was ensured by application of identical tank settings. All tests were run at a flow direction of −180° to the $x_t$ axis. No waves were generated and paddles were locked up, resulting in a tank depth of 2 m. The measurement period was defined as 2 min, longer than the stationarity period of the flow defined in Reference [21]. Changes made to tank nominal flow velocities were performed with sufficient time between changes for the flow to settle. Table 4 provides an overview of the flow speeds recorded by the SB-ADP along the tank profile and by the ADV at locations A,B,C and D. A dataset containing ADV measurements performed at the required locations for a nominal flow speed of

0.8 ms$^{-1}$ (impeller speed of 96 rpm), is openly available in Noble et al. [15]. The same instrument and a similar mounting technique as presented in this paper were used. Hence, this data was not reproduced in this dataset.

**Table 4.** Tank conditions measured by the SB-ADP and the ADV.

| Flow Speed (ms$^{-1}$) | Impeller Speed (rpm) | SB-ADP Profile | ADV | | | |
|:---:|:---:|:---:|:---:|:---:|:---:|:---:|
| | | | A | B | C | D |
| 0.6 | 73 | X | X | X | X | X |
| 0.8 | 96 | X | * | * | * | * |
| 1.0 | 120 | X | X | X | X | X |
| 1.2 | 144 | X | X | X | X | X |

* represents tank conditions that are available in [15]

## 3. Analysis

Quality control was conducted on the ADV and SB-ADP data. Prior to commencing data acquisition, the instruments were shown to be obtaining a high signal to noise ratio (SNR), a measure of the amplitude of the signal over the background noise level. A SNR above 15 dB is recommended for an ADV [17]. The SNR was not recorded by the SB-ADP but the amplitude, indicative of the strength of the backscattered signal, always exceeded 50 dB.

Each quality control process applied to the data generated a series of flags that were saved with the data-set (see Section 4). A despiking algorithm [22,23] was applied to all data using the phase-space method based on *Goring and Nikora* [24]. A correlation threshold of 70% was implemented, where correlation is a statistical measure of similarity in the received signal with respect to time. Data associated with correlations lower than 70% was flagged as failing quality control.

*SB-ADP: Anomaly Identification*

The amplitude of the returned signal typically decreases with distance from the instrument. Consequently, an amplitude increase with distance in one or more cells is indicative of an anomaly in the recorded data [16]. Anomalies were observed in some of the measurements acquired at lower tank flow velocities that is, 0.6 ms$^{-1}$ and 0.8 ms$^{-1}$. Figures 3 and 4 show two minute SB-ADP data-sets time averaged over each cell along the profile at 1.2 ms$^{-1}$ and 0.8 ms$^{-1}$ (Figure 3) and at 1.0 ms$^{-1}$ and 0.6 ms$^{-1}$ (Figure 4). The figures display (**a**) the velocity, (**b**) the amplitude of the returned signal and (**c**) the correlation in the received signal. Note that the instrument's convention is to record a negative flow speed for flow towards the device.

Anomalies were not observed in measurements acquired at tank nominal flow velocities of 1.2 ms$^{-1}$ and 1.0 ms$^{-1}$. At these tank velocities, mean measured velocity increases to its maximum 2.98 m from the SB-ADP transducer, at the tank centre, and decreases to the end of the measurement range. Mean amplitude of the returned signal decreases with increasing measurement range. Mean correlation between pulses increases from 89% at the first cell to plateau at approximately 96% at 5 m from the instrument. At tank nominal flow velocities of 0.8 ms$^{-1}$ and 0.6 ms$^{-1}$, anomalies appeared as steep gradient drops in velocity, as increases in the amplitude of the return signal with distance over an otherwise reducing amplitude profile and as increased scatter of the correlation.

The effect of the anomaly having more magnitude on the velocity data, measurement cells containing anomalous measurements have been flagged using a filter based on the rate of change in mean velocity between neighbouring measurement cells. The measurement cells, which have been flagged, are shown in red in Figures 3 and 4.

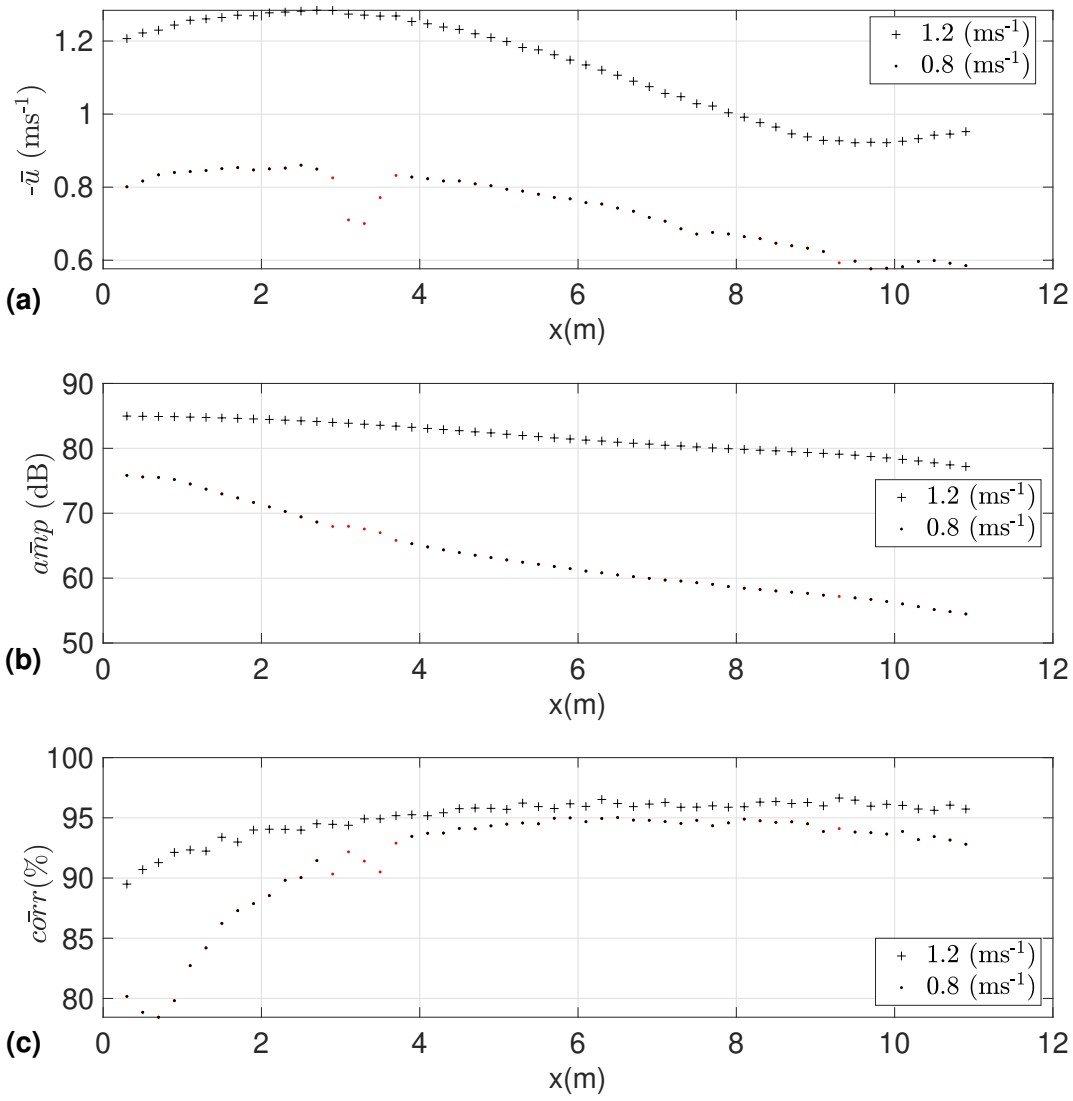

**Figure 3.** Cell temporal average of (**a**) velocity, (**b**) amplitude and (**c**) correlation after first quality control for a tank running at $0.8\ \mathrm{ms}^{-1}$ and $1.2\ \mathrm{ms}^{-1}$. Anomalies in the data-set are displayed in red.

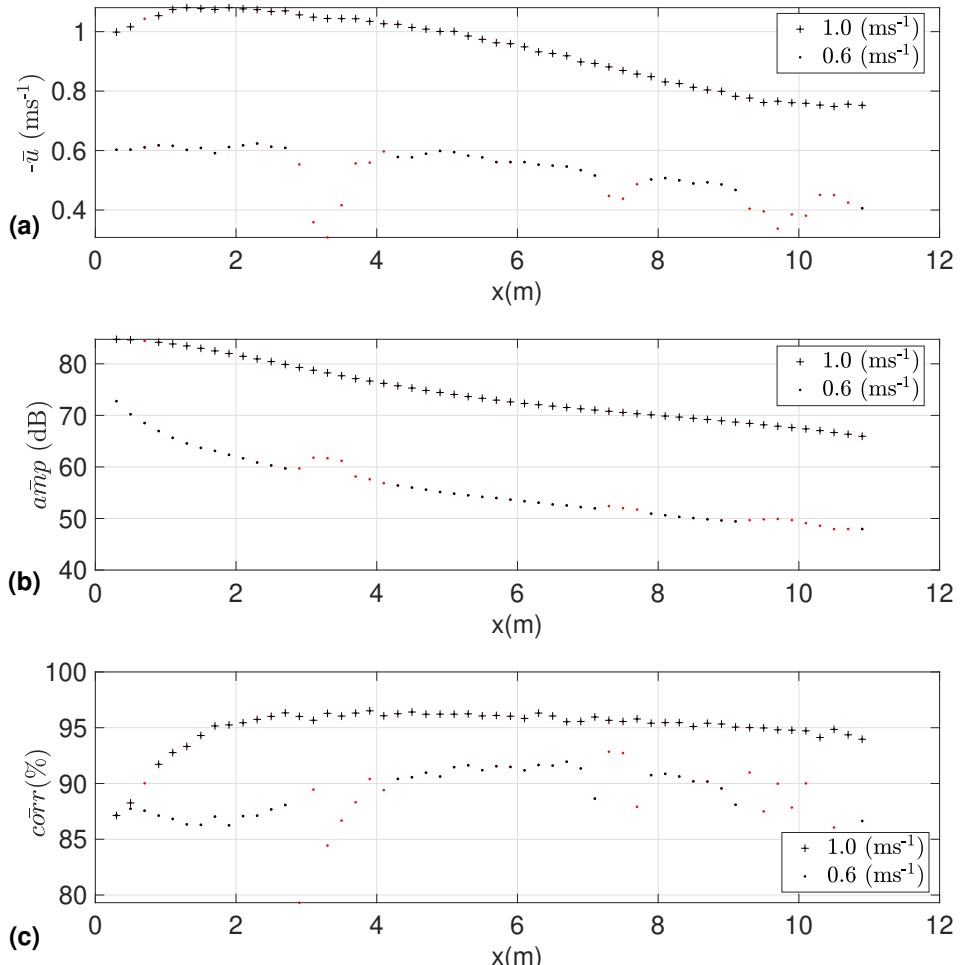

**Figure 4.** Cell temporal average of (**a**) velocity, (**b**) amplitude and (**c**) correlation after first quality control for a tank running at 0.6 ms$^{-1}$ and 1.0 ms$^{-1}$. Anomalies in the data-set are displayed in red.

## 4. Data

The data-set is available as a zip-file via the Datashare service of the University of Edinburgh [14]. It contains netCDF (Network Common Data Form) files, a machine independent format, free and open source [25].

The individual file naming convention is defined as follows:

ADV_[Flow speed][Location].nc
SBADP_[Flow speed].nc,

where "ADV" and "SBADP" are the type of instrument, and [Flow speed] and [Location] are as listed in Table 4.

The netCDF file contains metadata and data generated by the instruments. It also contains additional metadata describing the experimental conditions and quality control flags, making this dataset available for analysis. The structure is presented in Table 5, where * refers to characteristics exclusively applicable to the SB-ADP data file and ** refers to characteristics exclusively applicable to the ADV data file, otherwise the characteristics are applicable to both instrument files. If relevant, data dimensions are written in []; *Ndata* refers to the number of samples recorded during the 2 min measurement period and *Nbin* refers to the number of measurement cells (only relevant for the SB-ADP data).

Quality control flags have been added to each individual sample using the quality control processes described in Section 3. Flag values have been set to either 0 where the raw data point has passed the corresponding quality control, or 1 where the data point has failed quality control.

**Table 5.** Structure of the netCDF files containing metadata and data, where * refers to variables that only apply to the SB-ADP and ** only to the ADV.

| Instrument Specification | |
|---|---|
| Instrument type | |
| Model | |
| Firmware Version | |
| Number of beams | |
| Carrier Frequency (Hz) | |
| **Position** | |
| Test location | |
| Coordinates reference system | |
| X coordinates (m) | |
| Y Coordinates (m) | |
| Z Coordinates (m) | |
| **Clock** | |
| Time (s) BST | [*Ndata X 1*] |
| **Environment** | |
| Pressure (dbar) * | [*Ndata X 1*] |
| Pressure Offset * (dbar) | |
| Temperature (Celsius) | [*Ndata X 1*] |
| Salinity | |
| Speed Of Sound in water | [*Ndata X 1*] |
| **Tank settings** | |
| Tank nominal flow velocity (m/s) | |
| Tank speed setting (rpm) | |
| Tank flow direction | |
| Tank wave setting | |
| **Instrument orientation** | |
| Pitch * (°) | |
| Roll * (°) | |
| Heading * (°) | |
| Facing | |
| **Instrument configuration** | |
| Number of bins | |
| Bin size (m * or 0.1mm **) | |
| Bin start ** (0.1mm) | |
| Blanking distance * | |
| Sample rate (Hz) | |
| Transmitted power (dB) | |
| Probe calibration matrix ** | |

**Table 5.** *Cont.*

| Quality data | |
| --- | --- |
| Amplitude of the returned echo beam 1 (dB) | [*Ndata X nbin*] * or [*Ndata X 1*] ** |
| Ping to ping correlation of the returned echo beam 1 (%) | [*Ndata X nbin*] * or [*Ndata X 1*] ** |
| Signal to noise ration beam 1 ** (dB) | [*Ndata X 1*] ** |
| Amplitude of the returned echo beam 2 ** (dB) | [*Ndata X 1*] ** |
| Ping to ping correlation of the returned echo beam 2 ** (%) | [*Ndata X 1*] ** |
| Signal to noise ration beam 2 ** (dB) | [*Ndata X 1*] ** |
| Amplitude of the returned echo beam 3 ** (dB) | [*Ndata X 1*] ** |
| Ping to ping correlation of the returned echo beam 3 ** (%) | [*Ndata X 1*] ** |
| Signal to noise ration beam 3 ** (dB) | [*Ndata X 1*] ** |
| Amplitude of the returned echo beam 4 ** (dB) | [*Ndata X 1*] ** |
| Ping to ping correlation of the returned echo beam 4 ** (%) | [*Ndata X 1*] ** |
| Signal to noise ration beam 4 ** (dB) | [*Ndata X 1*] ** |
| **Velocity data** | |
| Velocity beam 1 * (m/s) | [*Ndata X nbin*] |
| Velocity along X ** (m/s) | [*Ndata X 1*] |
| Velocity along Y ** (m/s) | [*Ndata X 1*] |
| Velocity along Z1 ** (m/s) | [*Ndata X 1*] |
| Velocity along Z2 ** (m/s) | [*Ndata X 1*] |
| **Data quality flags** | |
| flag from phase-space threshold | [*Ndata X nbin*] * or [*Ndata X 1*] ** |
| flag correlation >70% | [*Ndata X nbin*] * or [*Ndata X 1*] ** |
| flag anomaly detected * | [*Ndata X nbin*] * or [*Ndata X 1*] ** |
| flag definition | |

## 5. Conclusions

This data provides multiple contemporaneous velocity measurements along the tank profile. This can be valuable for future testing in this specific facility as an empty tank baseline flow profile, validation of numerical models simulating flow hydrodynamics, or analysis of turbulent parameters. In addition, potential future SB-ADP users can analyse the provided dataset and conduct additional quality control and analysis particular to their specific application.

In the experimental set-up presented, ADV and SB-ADP measurements were performed individually due to the ADV set-up acting as a hard scatterer for the SB-ADP acoustic beam. The two systems could be used together if the ADV was located outwith the acoustic beam of the SB-ADP. This would provide multi-point measurement of uni-directional velocity along a profile, and point measurement of three-directional velocity at desired location(s). Future work is necessary to determine the necessary spatial separation of the SB-ADP and the ADV to obtain such measurements.

**Author Contributions:** Conceptualization, B.G.S., D.M.I., C.O., M.D., M.J.d.T. methodology, B.G.S., D.M.I., C.O., M.D., M.J.d.T.; formal analysis, M.J.d.T., B.G.S., D.M.I.; investigation, R.G.,B.G.S., C.O., M.D., M.J.d.T.; resources, T.D., R.G., B.G.S.; data curation, M.J.d.T., C.O.; writing, review and editing, (All, led by M.J.d.T.) ; visualization, M.J.d.T., B.G.S., D.M.I., C.O., M.D.; supervision, B.G.S., D.M.I.; project administration, T.D., B.S.G., D.M.I., R.G.; funding acquisition, D.M.I., T.D. All authors have read and agreed to the published version of the manuscript.

**Funding:** This research was funded by The UK Engineering and Physical Sciences Research Council through the Centre for Doctoral Training in Wind and Marine Energy System(EP/L016680/1). The single beam acoustic Doppler profiler was funded thanks to the European Commission's Horizon 2020 programme via the RealTide project (Grant Agreement no. 727689).

**Acknowledgments:** The authors wish to thank the staff of the FloWave Ocean Energy Research facility.

**Conflicts of Interest:** The authors declare no conflict of interest.

## Abbreviations

The following abbreviations are used in this manuscript:

| | |
|---|---|
| ADV | acoustic Doppler velocimeter |
| ADP | acoustic Doppler profiler |
| SB-ADP | single beam acoustic Doppler profiler |
| ORE | offshore renewable energy |
| LDV | laser Doppler velocimeter |
| SNR | signal to noise ratio |
| netCDF | Network Common Data Form |

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
