# Peer review of "Single-Beam Acoustic Doppler Profiler and Co-Located Acoustic Doppler Velocimeter Flow Velocity Data"

_data_

Round 1
Reviewer 1 Report
I think the idea of a comparison of measurements between a single beam ADP and an ADV is interesting.
The authors should provide a better justification for this research (why is it needed), what their original contribution is, and how it fits within the context of other similar work.
Results are presented for the ADP but not the ADV. Also missing is the ADP-ADV comparison. Isn't that the whole point of the paper?
The authors present an extensive description of the data format, but it is not clear what the significance of that is to a research publication.
I suggest the authors revise the paper to focus on the original scientific contributions, and provide comparison of data between the two instruments, along with error metrics.
Reviewer 2 Report
The paper "Single-beam Acoustic Doppler Profiler and co-located
Acoustic Doppler Velocimeter flow velocity data" describe correctly a database of measurements on the flow velocities in a tank, acquired with AVP and ADV.
The setup is described correctly and the same is for the file structure.
I suggest the authors to
- discuss any possible interference between the two acquisition systems, and
- discuss possible use of the acquied data in the conclusion.
Round 2
Reviewer 1 Report
Thanks to the authors for clarifying that the work presented is a supplemental description of the data, for the analysis presented in a Sensors manuscript.
My only suggestion at this point is that the above point be made abundantly clear in both the abstract and the introduction. This paper still reads as an independent body of work, which it seemingly is not.
Author Response
We thank the reviewer for his/her valuable work. Please see the response attached.
